# SELF-EDUCATED LANGUAGE AGENT WITH HINDSIGHT EXPERIENCE REPLAY FOR INSTRUCTION FOLLOWING

## ABSTRACT

Language creates a compact representation of the world and allows the description of unlimited situations and objectives through compositionality. These properties make it a natural fit to guide the training of interactive agents as it could ease recurrent challenges in Reinforcement Learning such as sample complexity, generalization, or multi-tasking. Yet, it remains an open-problem to relate language and RL in even simple instruction following scenarios. Current methods rely on expert demonstrations, auxiliary losses, or inductive biases in neural architectures. In this paper, we propose an orthogonal approach called Textual Hindsight Experience Replay (THER) that extends the Hindsight Experience Replay approach to the language setting. Whenever the agent does not fulfill its instruction, THER learns to output a new directive that matches the agent trajectory, and it relabels the episode with a positive reward. To do so, THER learns to map a state into an instruction by using past successful trajectories, which removes the need to have external expert interventions to relabel episodes as in vanilla HER.

## 1 INTRODUCTION

Language has slowly evolved to communicate intents, to state objectives, or to describe complex situations (Kirby et al., 2015). It conveys information compactly by relying on composition and highlighting salient facts. Such properties are essential when developing interactive agents in complex environments. As language may express a vast diversity of goals and situations, it could alleviate the training of interactive agents over heterogeneous and composite tasks thanks to its intrinsic structure (Luketina et al., 2019). This property is all the more critical as classic Reinforcement Learning (RL) methods are facing generalization issues (Cobbe et al., 2018), and learning hierarchical and structured policies remains an open-problem (Barto & Mahadevan, 2003; Kulkarni et al., 2016). As recently advocated by Luketina et al. (2019), language should thus be considered a first-class citizen to ease RL to improve on generalization and sample efficiency.

Unfortunately, conditioning a policy on language also entails a supplementary difficulty as the agent needs to understand linguistic cues to alter its behavior. The agent thus needs to ground its language understanding by relating the words to its observations, actions, and rewards before being able to leverage the language structure (Kiela et al., 2016; Hermann et al., 2017). Once the linguistic symbols are grounded, the agent may then take advantage of language compositionality to condition its policy on new goals. It thus leads to the following questions: are we eventually making the reinforcement learning problem harder, or can we generate learning synergies between policy learning and language acquisition?

In this work, we use instruction following as a natural testbed to examine this question (Tellex et al., 2011; Chen & Mooney, 2011; Artzi & Zettlemoyer, 2013; Luketina et al., 2019; Zang et al., 2018; Hermann et al., 2019; Chen et al., 2019). In this setting, the agent is given a text description of its goal (e.g. "pick the red ball") and is rewarded when achieving it. The agent has thus to visually grounded the language, i.e., linking and disentangling visual attributes (*shape*, *color*) from language description ("ball", "red") by using rewards to condition its policy toward task completion. On one side, the language compositionality allows for a high number of goals, and offers generalization opportunities; but on the other side, it dramatically complexifies the policy search space. Besides,

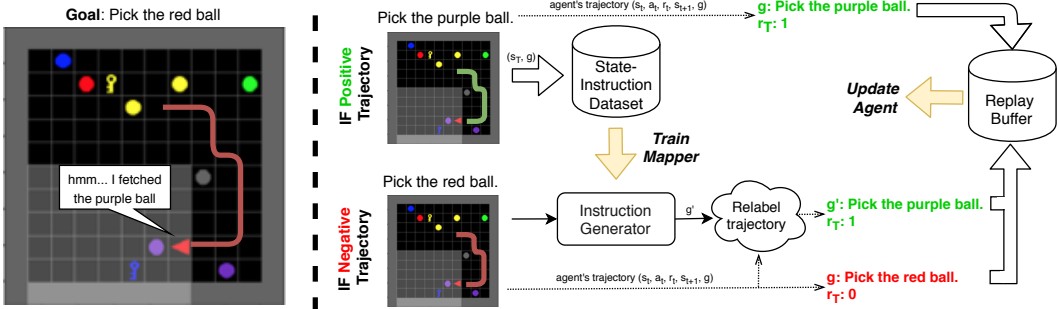

Figure 1: Upon positive trajectory, the agent trajectory is used to both update the RL replay buffer and the goal mapper training dataset. Upon failed trajectory, the goal mapper is used to relabel the episode, and both trajectories are appended to the replay buffer. In the original HER paper (Andrychowicz et al., 2017), the mapping function is bypassed since they are dealing with spatial goals, and therefore, vanilla HER cannot be applied without external feedback (Chan et al., 2018).

instruction following is a notoriously hard RL problem since it has a sparse reward signal. In practice, the navigation and language grounding problems are often circumvented by warm-starting the policy with labeled trajectories (Zang et al., 2018; Anderson et al., 2018). Although scalable, these approaches require numerous human demonstrations, whereas we here want to jointly learn the navigation policy and language understanding from scratch. In a seminal work, Hermann et al. (2017) successfully ground language instructions, but the authors used unsupervised losses and heavy curriculum to handle the sparse reward challenge.

In this paper, we take advantage of language compositionality to tackle the lack of reward signals. To do so, we extend Hindsight Experience Replay (HER) to language goals (Andrychowicz et al., 2017). HER originally deals with the sparse reward problems in spatial scenario; it relabels unsuccessful trajectories into successful ones by redefining the policy goal *a posteriori*. As a result, HER creates additional episodes with positive rewards and a more diverse set of goals. Unfortunately, this approach cannot be directly applied when dealing with linguistic goals. As HER requires a mapping between the agent trajectory and the goal to substitute, it requires expert supervision to describe failed episodes with words. Hence, this mapping should either be handcrafted with synthetic bots (Chan et al., 2018), or be learned from human demonstrations, which would both limit HER generality. More generally, language adds a level of semantics, which allows generating textual objective that could not be encoded simple spatial observations as in regular HER, e.g., "fetch a ball that is not blue" or "pick any red object".

In this work, we introduce Textual Hindsight Experience Replay (THER), a training procedure where the agent jointly learns the language-goal mapping and the navigation policy by solely interacting with the environment illustrated in Figure 1. THER leverages positive trajectories to learn a mapping function, and THER then tackles the sparse reward problem by relabeling language goals upon negative trajectories in a HER fashion. We evaluate our method on the BabyAI world (Chevalier-Boisvert et al., 2019), showing a clear improvement over RL baselines while highlighting the robustness of THER to noise.

## 2 BACKGROUND AND NOTATION

In reinforcement learning, an agent interacts with the environment to maximize its cumulative reward (Sutton & Barto, 2018). At each time step $t$, the agent is in a state $s_t \in \mathcal{S}$, where it selects an action $a_t \in \mathcal{A}$ according its policy $\pi : \mathcal{S} \rightarrow \mathcal{A}$. It then receives a reward $r_t$ from the environment's reward function $r : \mathcal{S} \times \mathcal{A} \rightarrow \mathbb{R}$ and moves to the next state $s_{t+1}$ with probability $p(s_{t+1}|s_t, a_t)$. The quality of the policy is assessed by the Q-function defined by $Q^\pi(s, a) = \mathbb{E}_\pi [\sum_t \gamma^t r(s_t, a_t)|s_0 = s, a_0 = a]$ for all $(s, a)$ where $\gamma \in [0, 1]$ is the discount factor. We define the optimal Q-value as $Q^*(s, a) = \max_\pi Q^\pi(s, a)$, from which the optimal policy $\pi^*$ is derived. We here use Deep Q-learning (DQN) to approximate the optimal Q-function with neu-

ral networks and perform off-policy updates by sampling transitions $(s_t, a_t, r_t, s_{t+1})$ from a replay buffer (Mnih et al., 2015).

In this article, we augment our environment with a goal space $\mathcal{G}$ which defines a new reward function $r : \mathcal{S} \times \mathcal{A} \times \mathcal{G} \rightarrow \mathbb{R}$ and policy $\pi : \mathcal{S} \times \mathcal{G} \rightarrow \mathcal{A}$ by conditioning them on a goal descriptor $g \in \mathcal{G}$. Similarly, the Q-function is also conditioned on the goal, and it is referred to as Universal Value Function Approximator (UVFA) (Schaul et al., 2015). This approach allows learning holistic policies that generalize over goals in addition to states at the expense of complexifying the training process. In this paper, we explore how language can be used for structuring the goal space, and how language composition eases generalization over unseen scenarios in a UVFA setting.

**Hindsight Experience Replay (HER)** (Andrychowicz et al., 2017) is designed to increase the sample efficiency of off-policy RL algorithms such as DQN in the goal-conditioning setting. It reduces the sparse reward problem by taking advantage of failed trajectories, relabelling them with new goals. An expert then assigns the goal that was achieved by the agent when performing its trajectory, before updating the agent memory replay buffer with an additional positive trajectory.

Formally, HER assumes the existence of a predicate $f : \mathcal{S} \times \mathcal{G} \rightarrow \{0, 1\}$ which encodes whether the agent in a state $s$ satisfies the goal $f(s, g) = 1$, and defines the reward function $r(s_t, a, g) = f(s_{t+1}, g)$. At the beginning of an episode, a goal $g$ is drawn from the space $\mathcal{G}$ of goals. At each time step $t$, the transition $(s_t, a_t, r_t, s_{t+1}, g)$ is stored in the DQN replay buffer, and at the end of an unsuccessful episode, an expert provides an additional goal $g'$ that matches the trajectory. New transitions $(s_t, a_t, r'_t, s_{t+1}, g')$ are thus added to the replay buffer for each time step $t$, where $r' = r(s_t, a_t, s_{t+1}, g')$. DQN update rule remains identical to (Mnih et al., 2015), transitions are sampled from the replay buffer, and the network is updated using one step td-error minimization.

HER assumes that a mapping $m$ between states $s$ and goals $g$ is given. In the original paper (Andrychowicz et al., 2017), this requirement is not restrictive as the goal space is a subset of the state space. Thus, the mapping $m$ is straightforward since any state along the trajectory can be used as a substitution goal. In the general case, the goal space differs from the state space, and the mapping function is generally unknown. In the instruction following setting, there is no obvious mapping from visual states to linguistic cues. It thus requires expert intervention to provide a new language goal given the trajectory, which drastically reduces the interest of HER. Therefore, we here explore how to learn this mapping without any form of expert knowledge nor supervision.

## 3 TEXTUAL HINDSIGHT EXPERIENCE REPLAY

Textual Hindsight Experience Replay (THER) aims to learn a mapping from past experiences that relates a trajectory to a goal in order to apply HER, even when no expert are available. The mapping function relabels unsuccessful trajectories by predicting a substitute goal $\hat{g}$ as an expert would do. The transitions are then appended to the replay buffer. This mapping learning is performed alongside agent policy training.

Besides, we wish to discard any form of expert supervision to learn this mapping as it would reduce the practicability of the approach. Therefore, the core idea is to use environment signals to retrieve training mapping pairs. Instinctively, in the sparse reward setting, trajectories with positive rewards encode ground-truth mapping pairs, while trajectories with negative rewards are mismatched pairs. These cues are thus collected to train the mapping function for THER in a supervised fashion. We emphasize that such signals are inherent to the environment, and an external expert does not provide them. In the following, we only keep positive pairs in order to train a discriminative mapping model.

Formally, THER is composed of a dataset $D$ of $\langle s, g \rangle$ pairs, a replay buffer $R$ and a parametrized mapping model $m_{\boldsymbol{w}}$. For each episode, a goal $g$ is picked, and the agent generates transitions $(s_t, a_t, r_t, s_{t+1}, g)$ that are appended to the replay buffer $R$. The Q-function parameters are updated with an off-policy algorithm by sampling minibatches from $D$. Upon episode termination, if the goal is achieved, i.e. $f(s_T, g) = 1$, the $\langle s_T, g \rangle$ pair is appended to the dataset $D$. If the goal is not achieved, a substitute goal is sampled from the mapping model[1] $m_{\boldsymbol{w}}(s_T) = \hat{g}'$ and the additional transitions $\{(s_t, a_t, r_t, s_{t+1}, \hat{g}')\}_{t=0}^{T}$ are added to the replay buffer. At regular intervals, the mapping

---

[1]The mapping model can be utilized with an accuracy criterion over a validation set to avoid random goal sampling. see algorithm 1 for more details

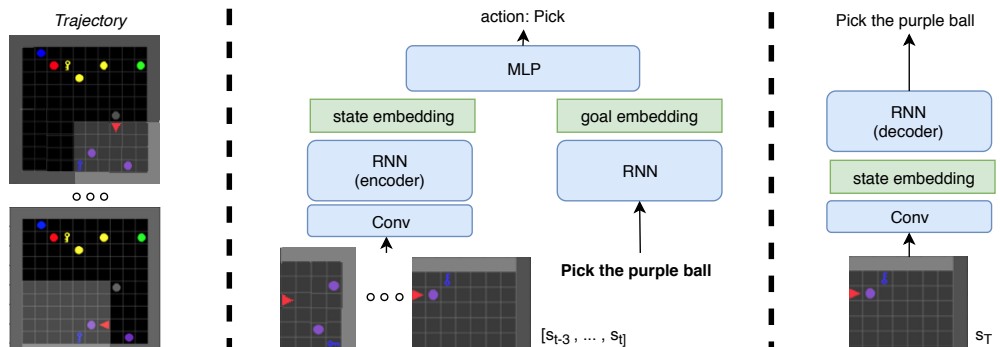

Figure 2: **Left**: Models are fed with the pre-extracted observations from the trajectories. **Middle**: Agent model whose inputs are the last four observations and the goal. **Right**: Instruction Generator.

model $m_{\boldsymbol{w}}$ is optimized to predict the goal $g$ given the trajectory $\tau$ by sampling mini-batches from $D$. Algorithm 1 summarizes our approach. Noticeably, THER can be extended to partially observable environments by replacing the predicate function $f(s, g)$ by $f(\tau, g)$, i.e., the completion of a goal depends on the full trajectory rather than one state. Although we assess THER in the instruction following setting, the proposed procedure can be extended to any other goal modalities.

# 4 EXPERIMENTS

## 4.1 EXPERIMENTAL SETTING

**Environment** We experiment our approach on a grid world environment called MiniGrid (Chevalier-Boisvert et al., 2019). This environment offers a variety of instruction-following tasks using a synthetic language for grounded language learning. We use a 10x10 grid with 10 objects randomly located in the room. Each object has 4 attributes (shade, size, color, and type) inducing a total of 300 different objects. The agent has four actions {forward, left, right, pick}, and it can only see the 7x7 grid in front of it. For each episode, one object's attribute is randomly picked as a goal, and the text generator translates it in synthetic language as detailed in Appendix C, e.g., "Fetch a tiny light blue ball." The agent is rewarded when picking one object matching the goal description, which ends the episode; otherwise, the episode stops after 40 steps or after taking an incorrect object.

**Models** In this experiment, THER is composed of two separate models as shown in Figure 2. The instruction generator is a neural network outputting a sequence of words given the final state of a trajectory. It is trained by gradient descent using a cross-entropy loss on the dataset $D$ collected as described in section 3. We train a DQN network following (Mnih et al., 2015) with a dueling head (Wang et al., 2016), double Q-learning (Hasselt et al., 2016), and a uniform replay buffer. The network receives a tuple $< s, g >$ as input and output an action corresponding to the argmax over states-actions values $Q(s, a, g)$. We use $\epsilon$-greedy exploration with decaying $\epsilon$. The detailed models and hyperparameters are provided in Appendix A, and the source code is available at HIDDEN_FOR_BLIND_REVIEW.

## 4.2 BUILDING INTUITION

This section examines the feasibility of THER by analysing two potential issues. We first show that HER is robust to a noisy mapping function (or partially incorrect goals), we then estimate the accuracy and generalisation performance of the instruction generator.

**Noisy instruction generator and HER** We investigate how a noisy mapping $m$ affects performance compared to a perfect mapping. As the learned instruction generator is likely to be imperfect, it is crucial to assess how a noisy mapping may alter the training of the agent. To do so, we train an agent with HER and a synthetic bot to relabel unsuccessful trajectories. We then inject noise

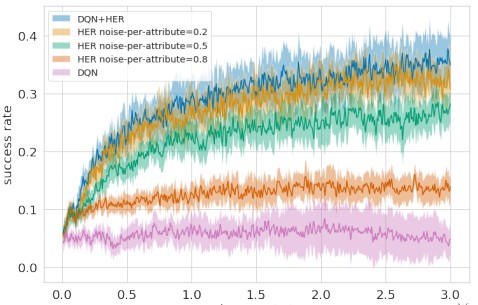 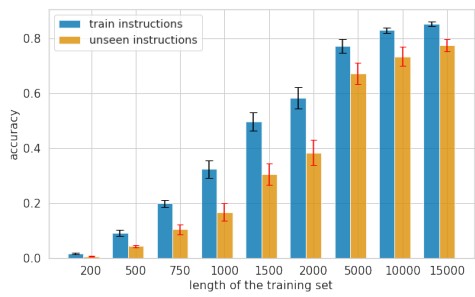

Figure 3: **Left**: Agent performance with noisy mapping function. **Right**: Instruction generator accuracy over 5k pairs. Figures are averaged over 5 seeds and error bars shows one standard deviation.

in our mapping where each attribute has a fixed probability $p$ to be swapped, e.g. color *blue* may be changed to *green*. For example, when $p = 0.2$, the probability of having the whole instruction correct is $0.8^4 \approx 0.4$. The resulting agent performance is depicted in Figure 3 (left).

The agent performs 80% as well as an agent with perfect expert feedback even when the mapping function has a 50% noise-ratio per attribute. Surprisingly, even highly noisy mappers, with a 80% noise-ratio, still provides an improvement over vanilla DQN-agents. Hence, HER can be applied even when relabelling trajectories with partially correct goals.

We also examine whether this robustness may be induced by the environment properties (e.g. attribute redundancy) rather than HER. We thus compute the number of discriminative features required to pick the correct object, as shown in Figure 8. On average, an object can be discriminated with 1.7 features in our setting - which eases the training, but any object shares at least one property with any other object 70% of the time - which tangles the training. Besides, the agent does not know which features are noisy or important. Thus, the agent still has to disentangle the instructions across trajectories in the replay buffer, and this process is still relatively robust to noise.

**Learning an instruction generator**    We briefly analyze the sample complexity and generalization properties of the instruction generator. If training the mapping function is more straightforward than learning the agent policy, then we can thus use it to speed up the navigation training.

We first split the set of missions $G$ into two disjoint sets $G_{train}$ and $G_{test}$. Although all object features are present in both sets, they contain dissimilar combinations of target objects. For instance, *blue*, *dark*, *key*, and *large* are individually present in instructions of $G_{train}$ and $G_{test}$ but the instruction to get a *large dark blue key* is only in $G_{test}$. We therefore assess whether a basic compositionality is learned. In the following, we use train/split ratio of 80/20, i.e., 240 vs 60 goals.

Finally, we generate an artificial dataset $D$ of $\langle g, s_T \rangle$ pairs, and we report the training/testing accuracy of the instruction generator in Figure 3 (right). The accuracy evaluates whether the four correct attributes are present in the linguistic instructions through a simple parser. For instance, "a large blue light key" is a failure case since one attribute is missing. Note that language accuracy is discussed further in subsection 4.4. Other language metrics can be used when dealing with natural language like BLEU (Papineni et al., 2002), ROUGE (Lin, 2004), METEOR (Banerjee & Lavie, 2005).

We here observe than 1000 positive episodes are necessary to reach around 20% accuracy with our model, and 5000 pairs are enough to reach 70% accuracy. The instruction generator also correctly predicts unseen instructions even with fewer than 1000 samples and the accuracy gap between seen and unseen instructions slowly decrease during training, showing basic compositionality acquisition. As further discussed in section 5, we here use a vanilla mapping architecture to assess the generality of our THER, and more advanced architectures may drastically improve sample complexity (Bahdanau et al., 2019b).

### 4.3    THER FOR INSTRUCTION FOLLOWING

In the previous section, we observe that: (1) HER is robust to noisy relabeled goals, (2) an instructor generator requires few positive samples to learn basic language compositionality. We thus here

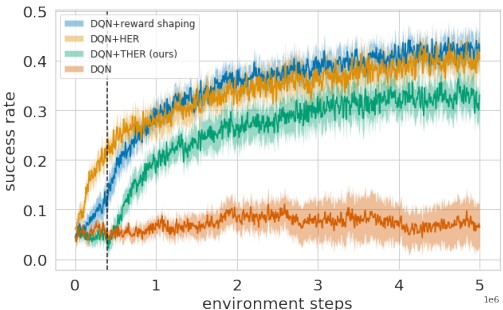 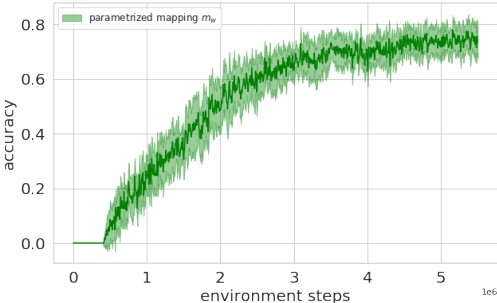

Figure 4: **Left**: learning curves for DQN, DQN+HER, DQN+THER in a 10x10 gridworld with 10 objects with 4 attributes. The instruction generator is used after the vertical bar. **Right**: the mapping accuracy for the prediction of instructions. $m_{w}$ starts being trained after collecting 1000 positive trajectories. Results are averaged over 5 seeds with one standard deviation.

combine those two properties to execute THER, i.e. jointly learning the agent policy and language prediction in a online fashion for instruction following.

**Baselines**  We want to assess if the agent benefits from learning an instruction generator and using it to substitute goals as done in HER. We denote this approach DQN+THER. We compare our approach to DQN without goal substitution (called DQN) and DQN with goal substitution from a perfect mapping provided by an external expert (called DQN+HER) available in the BabyAI environment. We emphasize again that it is impossible to have an external expert to apply HER in the general case. Therefore, DQN is a lower bound that we expect to outperform, whereas DQN+HER is the upper bound as the learned mapping can not outperform the expert. Note that we only start using the parametrized mapping function after collecting 1000 positive trajectories, which is around 18% validation accuracy. Finally, we compute an additional DQN baseline denoted DQN+reward: we reward the agent with 0.25 for each matching properties when picking a object given an instruction. It enforces a hand-crafted curriculum and dramatically reduces the reward sparsity, which gives a different perspective on the current task difficulty.

**Results**  In Figure 4 (left), we show the success rate of the benchmarked algorithms per environment steps. We first observe that DQN does not manage to learn a good policy, and its performance remains close to that of a random policy. On the other side, DQN+HER and DQN+reward quickly manage to pick the correct object 40% of the time. Finally, DQN+THER sees its success rates increasing as soon as we use the mapping function, to rapidly perform nearly as well as DQN+HER. Figure 4 (right) shows the performance accuracy of the mapping generator by environment steps. We observe a steady improvement of the accuracy during training before reaching 78% accuracy after 5M steps. In the end, DQN+THER outperforms DQN by using the exact same amount of information, and even matches the conceptual upper bond computed by DQN+HER. Besides, THER does not alter the optimal policy which can occur when reshaping the reward (Ng et al., 1999).

**Discussion**  As observed in the previous noisy-HER experiment, the policy success rate starts increasing even when the mapping accuracy is 20%, and DQN+THER becomes nearly as good as DQN+HER despite having a maximum mapping accuracy of 78%. It demonstrates that DQN+THER manages to trigger the policy learning by better leveraging environment signals compared to DQN. As the instruction generator focuses solely on grounding language, it quickly provides additional training signal to the agent, initiating the navigation learning process.

We observe that the number of positive trajectories needed to learn a non-random mapping $m_{w}$ is lower than the number of positive trajectories needed to obtain a valid policy with DQN (even after 5M environment steps the policy has 10% success rate). Noticeably, we artificially generate a dataset in section 4.2 to train the instruction generator, whereas we follow the agent policy to collect the dataset, which is a more realistic setting. For instance, as the instructor generator is trained on a moving dataset, it could overfit to the first positive samples, but in practice it escapes from local minima and obtains a final high accuracy.

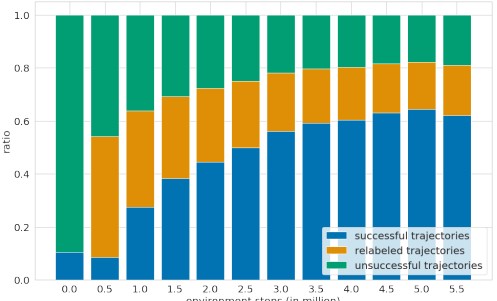 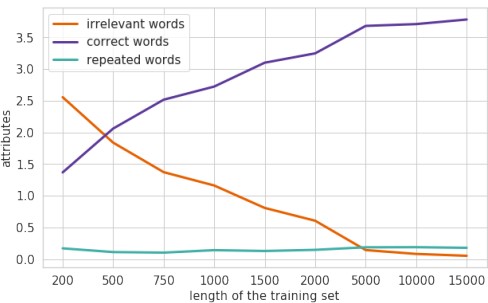

Figure 5: **Left**: Transition distributions in the replay buffer between successful, unsuccessful and relabeled trajectories. We remove time-out trajectories for clarity, which accounts for $54\%$ of the transition in average ($\pm 3\%$ over training). **Right**: Evaluating the language learned by the instruction generator on unseen instructions. Over time, the number of correct attributes (purple) is increasing, as the number of irrelevant words (orange) is decreasing. The number of repeated attributes (green) stays low. The beginning clause is ignored as it doesn't provide information regarding the objective.

Different factors may also explain the learning speed discrepancy: supervised learning has less variance than reinforcement learning as it has no long-term dependency. The agent instructor generator can also rely on simpler neural architectures than the agent. Although THER thus takes advantage of those training facilities to reward the agent ultimately.

Finally, we observe a virtuous circle that arises. As soon as the mapping is correct, the agent success rate increases, initiating the synergy. The agent then provides additional ground-truth mapping pairs, which increases the mapping accuracy, which improves the quality of substitute goals, which increases the agent success rate further more. As a result, there is a natural synergy that occurs between language grounding and navigation policy as each module iteratively provides better training samples to the other model. This virtuous circle is observed inside the replay buffer distribution, as shown in Figure 5. If we ignore time-out trajectories, around 90% of the trajectories are negative at the beginning of the training. As soon as we start using the instruction generator, 40% the transitions are relabelled by the instructor generator, and 10% of the transitions belong to positive trajectories. As training goes, this ratio is slowly inverted, and after 5M steps, there is only 15% relabelled trajectories left while 60% are actual positive trajectories.

**Limitations** Albeit generic, THER also faces some inherent limitations. From a linguistic perspective, THER cannot transcribe negative instructions (*Do not pick the red ball*), or alternatives (*Pick the red ball or the blue key*) in its current form. However, this problem could be alleviated by batching several trajectories with the same goal. Therefore, the model would potentially learn to factorize trajectories into a single language objective. On the policy side, THER still requires a few trajectories to work, and it thus relies on the navigation policy. In other word, historical HER could be applied in the absence of reward signals, while THER only alleviate the sparse reward problem by better leveraging successful trajectories. A natural improvement would be to couple THER with other exploration methods, e.g, intrinsic motivation (Bellemare et al., 2016) or DQN with human demonstration (Hester et al., 2018). Finally, under-trained goal generators might hurt the training in some environments although we did not observed it in our setting as shown in Figure 9. However, a simple validation accuracy allows to circumvent this risk while activating the goal mapper (More details in algorithm 1). We emphasize again that the instruction generator can be triggered anytime to kick-start the learning as the it is independent of the agent.

## 4.4 LANGUAGE LEARNED BY THE INSTRUCTION GENERATOR

We here analyze further the language quality of the instruction generator. To do so, we rely on three metrics to assess the generated language quality. The first metric, called *attribute fidelity*, assesses whether every target attribute is present in the generated sentence. For example, for the objective *a large dark blue key*, the generated sentence "Fetch me a large key" only containing two attributes and receives a score of two. Yet, the model may still enumerate all available attributes in a single sentence; the score would always be four. *Language precision* counter this effect by counting how

| Instruction #Samples | get a small very_light green key | get a tiny dark yellow key | go fetch a dark grey giant ball |
|---|---|---|---|
| 200 | get a neutral very_light tiny ball | get a small blue dark ball ball | go get a grey giant neutral giant neutral grey |
| 1000 | get a very_light green small ball | go fetch a tiny dark tiny ball | you must fetch a grey dark giant ball |
| 10000 | get a very_light green small key | go get a dark yellow tiny key | go fetch a grey dark giant ball |

Table 1: Examples of language errors during the training

many words are not relevant to describe the target object. This second metric is related to precision (or positive predictive value), as generated instructions only contain relevant attributes. Finally, we count *repeated attributes* as language models are known to stutter during early training.

In Figure 5, we compute the three metrics over unseen goal states, examining the compositionality properties of the instruction generator. We observe that generated instructions get more accurate, contain less irrelevant attributes, thus providing the agent with valid goals, even in unseen scenarios. As the instruction generator is trained until convergence as new <state, instruction> pairs are collected, it naturally preserve the overall language structure, and correctly ground symbols: *repeated attributes* score remains low and generated sentences start with the verb and end with the noun while randomly shuffling the attributes as shown in Table 1.

## 5 RELATED WORK

Instruction following have recently drawn a lot of attention following the emergence of several 2D and 3D environments (Chevalier-Boisvert et al., 2019; Brodeur et al., 2017; Anderson et al., 2018). This section first provides an overview of the different approaches, i.e, fully-supervised agent, reward shaping, auxiliary losses, before exploring approaches related to THER.

**Vision and Language Navigation** Instruction following is sometimes coined as *Vision and Language Navigation* tasks in computer vision (Anderson et al., 2018; Wang et al., 2019). Most strategies are based on imitation learning, relying on expert demonstrations and knowledge from the environment. For example, Zang et al. (2018) relate instructions to an environment graph, requiring both demonstrations and high-level navigation information. Closer to our work, Fried et al. (2018) also learns a navigation model and an instruction generator, but the latter is used to generate additional training data for the agent. The setup is hence fully supervised, and requires human demonstrations. These policies are sometimes finetuned to improve navigation abilities in unknown environments. Noticeably, Wang et al. (2019) optimizes their agent to find the shortest path by leveraging language information. The agent learns an instruction generator, and they derive an intrinsic reward by aligning the generator predictions over the ground truth instructions. Those approaches complete long sequences of instructions in visually rich environments but they require a substantial amount of annotated data. In this paper, we intend to discard human supervision to explore learning synergies. Besides, we needed a synthetic environments with experts to evaluate THER. Yet, THER could be studied on natural and visually rich settings by warm-starting the instruction generator, and those two papers give a hint that THER could scale up to larger environment.

**IRL for instruction following** Bahdanau et al. (2019a)learn a mapping from <instruction, state> to a reward function. The method's aim is to substitute the environment's reward function when instructions can be satisfied by a great diversity of states, making hand-designing reward function tedious. Similarly, Fu et al. (2019) directly learn a reward function and assess its transferability to new environments. Those methods are complementary to ours as they seek to transfer reward function to new environment and we are interested in reducing sample complexity.

**Improving language compositionality** THER heavily relies on leveraging the language structure in the instruction mapper toward initiating the learning synergy. For instance, Bahdanau et al. (2019b) explore the generalization abilities of various neural architectures. They show that the sample efficiency of feature concatenation can be considerably improved by using feature-wise modulation (Perez et al., 2018), neural module networks (Andreas et al., 2016) or compositional attention networks (Hudson & Manning, 2018). In this spirit, Bahdanau et al. (2019a) take advantage of these architectures to quickly learn a dense reward model from a few human demonstrations in the instruc-

tion following setup. Differently, the instructor generator can also be fused with the agent model to act as an auxiliary loss, reducing further the sparse reward issue.

**HER variants**    HER has been extended to multiple settings since the original paper. These extensions deal with automatic curriculum learning (Liu et al., 2019), dynamic goals (Fang et al., 2019), or they adapt goal relabelling to policy gradient methods (Rauber et al., 2019). Closer to our work, Sahni et al. (2019) train a generative adversarial network to hallucinate visual near-goals state over failed trajectories. However, their method requires heavy engineering as visual goals are extremely complex to generate, and they lack the compact generalization opportunities inherent to language. Chan et al. (2018) also studies HER in the language setting, but the authors only consider the context where a language expert is available.

**Conditioned Language Policy**    There have been other attempts to leverage language instruction to improve the agent policy. For instance, Jiang et al. (2019) computes a high-level language policy to give textual instruction to a low-level policy, enforcing a hierarchical learning training. The authors manage to resolve complicated manipulating task by decomposing the action with language operation. Yet, the language mapper performs instruction retrieval into a predefined set of textual goals, which prevent from benefiting from language compositionality, as mentioned by the authors. Co-Reyes et al. (2018) train an agent to refine its policy by collecting language corrections over multiple trajectories on the same task. While the authors focus their effort on integrating language cues, it could be promising to learn the correction function in a THER fashion.

## 6    CONCLUSION

We introduce Textual Hindsight Experience Replay (THER) as an extension to HER for language. We define a protocol to learn a mapping function to relabel unsuccessful trajectories with predicted consistent language instructions. We show that THER nearly matches HER performances despite only relying on signals from the environment. We provide empirical evidence that THER manages to alleviate the instruction following task by jointly learning language grounding and navigation policy with training synergies. THER has mild underlying assumptions, and it does not require human data, making it valuable to complement to other instruction following methods. More generally, THER can be extended to any goal modalities, and we expect similar procedures to emerge in other setting.

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

# A APPENDIX A: TRAINING DETAILS

## A.1 INSTRUCTION GENERATOR ARCHITECTURE

**The encoder** is a convolution neural network (LeCun et al., 1995) with ReLU activation functions after each layer for processing an observation of dimension 7x7x3. It is composed of: 16 of 2x2 kernel, followed a max pooling 2D of size 2x2 then 32 of 2x2 kernel, and finally 256 of 2x2 kernel. For the convolutional layers and the pooling layer, the stride is always equal to 1.

**The decoder** is a recurrent neural network composed of gated recurrent units (GRU) (Chung et al., 2014) that outputs the instruction word by word. The convolutional layers extract relevant information from the image and compress it in a latent representation. This latent representation is then used as the initial hidden state of a GRU layer. The initial input of the GRU layer is the token *BEGIN*. At each time step, the GRU layer outputs a distribution over words as in Seq2Seq (Sutskever et al., 2014), we can use as the input of the next step the word the is the most likely. When the token *END* is chosen the prediction stops. We use a word embedding of dimension 128 and an instruction embedding of dimension 256. The number of words in the dictionary is equal to 27.

The training is done with cross-entropy between the distribution of probabilities over words predicted by the model and the true word. Teacher forcing is employed to accelerate and to stabilize the learning. Teacher forcing refers to using the ground truth for the next input of the GRU instead of using the last predicted word (Bengio et al., 2015).

The learning rate is fixed to $10^{-4}$ and the network is trained using the Adam optimizer with default parameters (Kingma & Ba, 2015) and a regularization of $10^{-6}$ over all parameters. The batchsize is equal to 128.

## A.2 DQN ARCHITECTURE

To deal with the partial observability of the environment (described in subsection 4.1) the state corresponds to the last 4 frames stacked as in (Mnih et al., 2015).

**Visual Encoder**: Each frame is encoded by a convolutional neural network and then passed through a LSTM. Each layer is followed by a ReLU activation function. The convolutional neural network is composed of: 16 of 2x2 kernel, followed a max pooling 2D of size 2x2 then 32 of 2x2 kernel, and finally 64 of 2x2 kernel. For the convolutional layers and the pooling layer, the stride is always equal to 1. The LSTM has an input and hidden size of 64. The last LSTM hidden state $h_t$ corresponds to visual embedding.

**Instruction Encoding**: First each word is embedded with an embedding of size 32. Instructions are encoded word by word using a GRU. The GRU has an input size of 32 which corresponds to the word embedding size and a hidden size of 128 which corresponds to the instruction embedding size. The last GRU hidden state $h_t$ is kept as the instruction embedding.

We concatenate the visual embedding and the instruction embedding and add fully connected layers in the same fashion as in the dueling architecture (Wang et al., 2016) on top to compute the Q-values for each action.

**Training parameters** The exploration policy is $\epsilon$-greedy with $\epsilon$ decaying linearly from 1 to 0.05 either in 500 000 steps for DQN and DQN+THER or in 100 000 steps for DQN+HER. The RMSprop optimizer is used to train the neural network with a learning rate fixed to $10^{-5}$ and default parameters (the forgetting factor is 0.99). Huber loss (with $\delta = 1$) and gradient clipping (between -1 and 1) are used for stable gradients. The target network is synchronized with the current model every 1000 gradient steps. The replay buffer size is 50 000.

# B  THER ALGORITHM DETAILED

---

**Algorithm 1:** Textual Hindsight Experience Replay (THER)

---

**Given:**

- an off-policy RL algorithm (e.g. DQN) $\mathbb{A}$
- a reward function $r : \mathcal{S} \times \mathcal{A} \times \mathcal{G} \rightarrow \mathbb{R}$.
- a language score (e.g. parser accuracy, BLEU etc.)

1 Initialize $\mathbb{A}$ , replay buffer $R$, dataset $D_{train}$ and $D_{val}$ of $\langle instruction, state \rangle$, Instruction Generator $m_{\boldsymbol{w}}$;
2 **for** *episode=1,M* **do**
3      Sample a goal $g$ and an initial state $s_0$;
4      $t = 0$;
5      **repeat**
6          Execute an action $a_t$ chosen from the behavioral policy $\mathbb{A}$: $a_t \leftarrow \pi(s_t || g)$;
7          Observe a reward $r_t = r(s_t, a_t, g)$ and a new state $s_{t+1}$;
8          Store the transition $(s_t, a_t, r_t, s_{t+1}, g)$ in $R$;
9          Update Q-network parameters using the policy $\mathbb{A}$ and sampled minibatches from $R$;
10          $t = t + 1$;
11      **until** *episode ends*;
12      **if** $f(s_t, g) = 1$ **then**
13          Store the pair $\langle s_t, g \rangle$ in $D_{train}$ or $D_{val}$;
14          Update $m$-network parameters by sampling minibatches from $D_t rain$;
15      **end**
16      **else**
17          **if** *m language validation score is high enough and $D_{val}$ is big enough* **then**
18              Sample $\hat{g}' = m_{\boldsymbol{w}}(s_t)$;
19              Replace $g$ by $\hat{g}'$ in the transitions of the last episode and set $\hat{r} = r(s_t, a_t, \hat{g}')$.
20          **end**
21      **end**
22 **end**

---

## C  APPENDIX B: ENVIRONMENT DETAILS

A screenshot of the environment is provided in Figure 6.

The state of the environment does not correspond to a RBG image but to channels encoding different info about each cell (color, type etc...). More details are available at gym-minigrid.

The synthetic language used for instructions is composed in three parts. First, a clause like *get a* (all the clauses are displayed below). Then, a series of attributes randomly ordered describing the object and lastly, the type of the object. An example of instruction is *Go fetch a tiny dark red ball*. One object can be described by a maximum of 4 attributes: shade, size, color, and type. They have modalities going from 2 to 6.

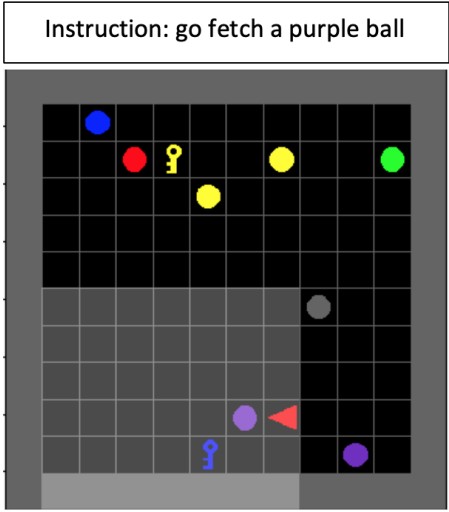

Figure 6: A screenshot of the environment. The agent only sees the light gray area. It should be noted that in is this example, object only have 2 attributes (color and type) but in our experiments, 4 attributes are being used.

The environment used is a grid-world of variable size containing objects. Objects can be described using up to 4 attributes. The list of all attributes is the following:

- **Shapes** *key*, *ball*
- **Colors** *red*, *green*, *blue*, *purple*, *yellow*, *grey*
- **Shades** *very_dark*, *dark*, *neutral*, *light*, *very_light*
- **Sizes** *tiny*, *small*, *medium*, *large*, *giant*

The five possible clauses are:

- *get a*
- *go fetch a*
- *go get a*
- *fetch a*
- *you must fetch a*

Multiple possible clauses bring diversity to the language as their meaning is equivalent.

## D  COMPLEMENTARY EXPERIMENT

**N-gram measure**  An n-gram is a sequence of words, e.g. 2-gram corresponds to a two-word sequence. For example the sentence *Get a red ball.* is composed of three 2-gram: *Get a*, *a red*, *red ball* and two 3-gram: *get a red* and *a red ball*. The n-gram measure assesses the language model accuracy by counting how many n-grams in the original sentence is present in the generated one. This measure is close to BLEU score used in machine translation (Papineni et al., 2002).

In our experiments, the language used is synthetic, and attributes *order* is random. Therefore, the attributes' presence is more important than the position of each word. To assess the learned language accuracy, we compare generated sequences to what we call *randomized ground truth*. Comparing generated instruction to instructions generated by the environment is not relevant as the beginning clause (i.e., *Get a* or *Fetch a*, etc.) and attributes order are random. Therefore, for a given ground truth instruction, object attributes are shuffled in the sentence, and the beginning clause is sampled from all possible clauses. Since the beginning clause is random, even randomized ground truth cannot reach an accuracy of 1. The lower bound called *Random Attributes* corresponds to sampling a clause and each attribute randomly.

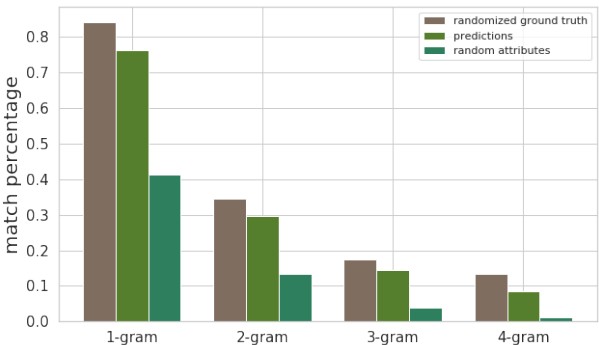

Figure 7: Quality of the language learned by the instruction generator with 10 000 samples. For *randomized ground truth* the sentences are the same as the ones from the ground truth but the order of the attributes is shuffled and the clause is changed. For *random attributes* beginning clause and object attributes are picked randomly.

Figure 7 shows that the language learned by the instruction generator is close to the upper bound *randomized ground truth*. These results correlate with Figure 5 (right), indicating that the instruction generator can produce instruction containing correct object attributes. The fast decrease in accuracy when *n* grows can be explained by the attributes order randomness.

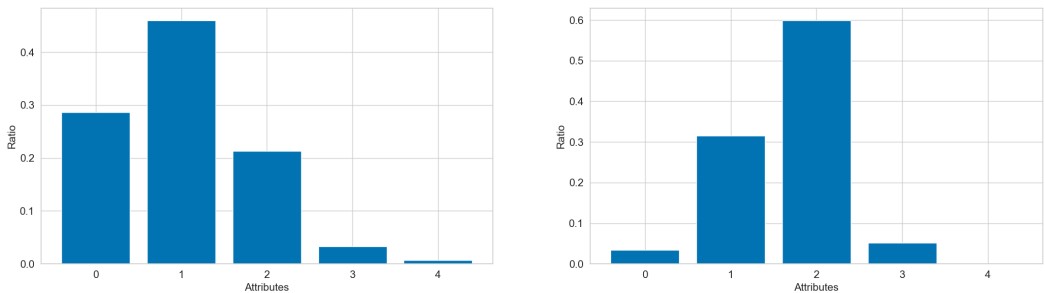

Figure 8: **Left**: Number of shared attributes between two objects. **Right**: The number of attributes needed to discriminate an object from all the others.

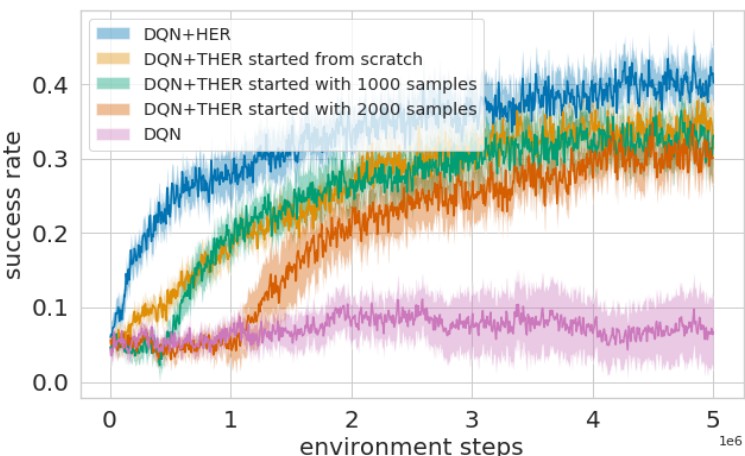

Figure 9: The instruction generator is triggered after collecting 0, 1000 and 2000 positive trajectories, i.e, approximately 0%, 20%, 50% accuracy following section 4.2. Even when the instruction generator is not accurate, the policy still makes steady progress and the final success rate is not impacted. Delaying the generator instructor does not provide additional benefit

