# OpenReview forum: "Self-Educated Language Agent with Hindsight Experience Replay for Instruction Following"
_ICLR.cc/2020/Conference — Reject_

### Official Review · AnonReviewer2 · 2019-10-07
**Official Blind Review #2**

**Rating:** 6

**Review:**

This paper proposes THER (textual hindsight experience replay), which extends the HER algorithm to the case when goals are represented as language. Whereas in HER the mapping from states to goals is the identity function, THER trains a separate modeling network which performs a mapping from states to goals represented by language. The policy (represented as a Q-function trained via DQN) takes in the goal (a command) as an additional argument as done in HER, which allows the agent to be commanded different tasks. The authors evaluate THER on the MiniGrid environment, where they demonstrate that THER greatly outperforms vanilla goal-conditioned DQN, even in the presence of significant label noise.

Overall, combining HER with language-based goals is an interesting and novel problem, and potentially a promising approach to solving language-conditioned reinforcement learning where sparse rewards are common. The authors show fairly convincingly that THER heavily outperforms DQN, which fails to improve from the random initial policy. However, I have several conceptual concerns with the proposed algorithm:

1) There seems to be a bootstrapping problem in the algorithm, with regards to the instruction generator and the policy. If the algorithm does not succeed in reaching goals, then the instruction generator m_w has little training data. However, if m_w is not good, then the algorithm will not be able to give good reward signal to the policy. HER does not have this problem as it has an oracle goal mapping function, so m_w is always good. Evidently, the algorithm worked on the domain that it was tested in, but do the authors have any intuition on when this bootstrapping behavior could be harmful, or some justification on why it would not happen? If the language was more complex (and not limited to a small set of template instructions), would the THER approach still be reasonable?

2) How does the algorithm detect if a given goal state corresponds to successful execution of a command? Or in the notation of the paper, how is f(s,g) implemented? In general, this does not seem like a trivial question to answer if one were to implement this algorithm in a real-world scenario.

My overall decision is borderline (learning towards accept), as the experiments were well done and serve as a good proof-of-concept, but I am unsure if this approach will scale well outside of the particular tested domain.

**Experience Assessment:**

I have published one or two papers in this area.

**Review Assessment: Checking Correctness Of Derivations And Theory:**

N/A

**Review Assessment: Checking Correctness Of Experiments:**

I carefully checked the experiments.

**Review Assessment: Thoroughness In Paper Reading:**

I read the paper thoroughly.

---

> ### Author Response · Authors · 2019-11-14
> **Official Comment**
>
> First of all, thank you for reading the paper thoroughly and taking the time to review it.
>
>
> 1) As you mentioned, there is a potential bootstrapping problem when using incorrect mapping (Reviewer #3 speaks about a negative feedback loop). If the mapping is not functional, we can fear that the policy cannot learn the correct behavior, or start degenerating. One of the paper contributions was to observe that this situation does not occur in our experiments (even in a highly noisy setting). Following the rebuttal, we also run THER by triggering the mapper from the beginning to invalid trajectories in our buffer. Yet, we did not observe a substantial loose of performance (cf Appendix).
>
> One potential intuition would be the following: if the instructions are invalid, the agent would follow an "average" policy, e.g., pick random objects; the goal-space would either be ignored or turned into a random representation. As those are non-degenerate cases, the agent may quickly recover as soon as the mapper becomes partially correct. Therefore, THER may only slow down the training in the worst-case scenario.
> We also want to emphasize that THER can be activated any time during the training, and a simple solution consists of waiting until the mapper reached an acceptable performance. In this spirit, we updated the paper to explicitly assess the mapper quality by using a validation accuracy, and triggering it once a threshold is reached (using only collected samples, no external data is necessary to validate the generator, cf. Algorithm in the Appendix). Besides, we can use any language metric (BLEU, RED, SPiCE, METEOR etc.) to replace the parser accuracy (We updated Section 4.2, second paragraph). Therefore, we could automatically assess a minimum level of language generation before triggering the instructor generator. In simple cases, the generator is learnt quickly and can be used right from the start and in harder setup, the generator kicks in when the validation accuracy is high enough, avoiding potential mislabeling at early stages.
>
>
> If the language was more complex, the goal generator might not be able to learn the language from scratch. However, we can ease language learning by using pretrained word embedding, e.g. GLoVe [1] or Fasttext [2], or using annotated trajectory. For instance, [3] and [4] successfully train an instruction generator in the Room2Room dataset with only 21k instructions.
>
>
> "HER does not have this problem as it has an oracle goal mapping function,"
> -> This is a fair point that we may not have emphasized enough in the paper: HER works in the absence of signals while THER only alleviate the sparse reward problem setting. We thus make it explicit in the limitation paragraph.
>
>
> 2) In the UFVA MDP setting, $f(s,g)$ is  given by the environment [5,6]. More precisely, the terminal state is an absorbing state whose transition entails the final reward. In practice, we can use the reward function as a stopping criterion, and in this paper, we implement it as follow $f(g,s) = r$. In a more realistic scenario, f is often hard-coded, e.g. the agent is close enough from the objective point. There has also been emergent literature in vision-and-language navigation tasks to tackle this issue [7,8]. For instance, the agent has to learn when to stop to answer questions on his environment in embodied question answering [9]. In those scenarios, an additional stopping module is learned from data.
>
>
> Again, thank you very much for your comments, we hope to have answered some of your concerns such as the UFVA setting, invalid bootstrapping, and a practical solution to handle natural language. We are open to discussion or additional changes if you think they can improve the paper further.

---

> > ### Author Response · Authors · 2019-11-14
> > **References**
> >
> >
> > [1] Pennington, Jeffrey, Richard Socher, and Christopher Manning. "Glove: Global vectors for word representation." Proceedings of the 2014 conference on empirical methods in natural language processing (EMNLP). 2014.
> > [2] https://github.com/facebookresearch/fastText
> > [3] Wang, Xin, et al. "Reinforced cross-modal matching and self-supervised imitation learning for vision-language navigation." Proceedings of the IEEE Conference on Computer Vision and Pattern Recognition. 2019.
> > [4] Fried, Daniel, et al. "Speaker-follower models for vision-and-language navigation." Advances in Neural Information Processing Systems. 2018.
> > [5] Schaul, Tom, et al. "Universal value function approximators." International Conference on Machine Learning. 2015.
> > [6] Andrychowicz, Marcin, et al. "Hindsight experience replay." Advances in Neural Information Processing Systems. 2017.
> > [7] Chen, Howard, et al. "Touchdown: Natural language navigation and spatial reasoning in visual street environments." Proceedings of the IEEE Conference on Computer Vision and Pattern Recognition. 2019.
> > [8] Das, Abhishek, et al. "Neural modular control for embodied question answering." arXiv preprint arXiv:1810.11181 (2018).
> > [9] Jiannan Xiang et al. "Not All Actions Are Equal: Learning to Stop in Language-Grounded Urban Navigation." Visually Grounded Interaction and Language Workshop. 2019

---

### Official Review · AnonReviewer3 · 2019-10-22
**Official Blind Review #3**

**Rating:** 6

**Review:**

This work attempts to learn instruction following agents without the requirement for a paired instruction, behavior corpus and instead simply relying on environment interaction and using a form of hindsight relabeling to learn how to relate language and behavior.

Introduction:

It thus leads to the following questions: are we eventually making the rein-forcement learning problem harder, or can we generate learning synergies between policy learningand language acquisition? -> it really doesn’t seem like the point of instruction following is that. It seems like you want instruction following so that you can communicate novel objectives to your agent, with the promise of generalization. Maybe reword?

The motivation for using HER makes sense, but maybe a bit more would be useful to describe why we need text here and not just regular HER.

I think this line “triggers alearning synergy between language acquisition and policy learning” is pretty confusing and not really adding too much value. Would remove.

Overall motivation makes sense, do language grounding in an interactive way much more effectively by leveraging hindsight relabeling but to get around the circular problem of hindsight generation leverage a model of successful behaviors seen thus far. This is a pretty neat thing to do!

Textual HER:
“We emphasize that such signals are inherent to the environment, and an external expert does not provide them” -> I do not think this is true. Rewards do not magically show up in the environment, they have to be provided. I get what you’re trying to say but this statement is very often not true. Please revise.

Conceptual question: what happens if none of the random trajectories are successful coz the reward is so sparse? Wouldn’t this be prohibitive? Importantly, I would be curious to understand how the number of entries in D affects the relabeling function m_omega and how this can be good or bad depending on the schedule of training. For instance if the D is very small at the start, it’s not going to be very good at doing the relabeling and might be erroneous.

Experiments

“Surprisingly, even highly noisy mappers, with a 80%noise-ratio, still provides an improvement over vanilla DQN-agents” -> do you know why?

The synthetic noisiness that you introduce is from a different distribution than the type of noisiness you’d expect from just having very few successful trajectories to train m_omega right? How can we evaluate that?

If we consider the number of successful trajectories obtained just by accident, is it even 1000 or 5000 as required to train the m_omega. If this is a negative feedback loop couldn’t it just keep getting worse because we are using erroneous m. Should we use some notion of uncertainty or something to know when to relabel with m? Or does it happen always

I generally like Section 4.2 -> nicely motivated!

“We emphasize again that it is impossible to have an external expert to apply HER in the general case” -> why??

Can the authors introduce other baselines. For instance the recent paper from Yiding Jiang, Chelsea Finn, Shane Gu and others might be a start. Consider corpus based instruction following would be another. Maybe these can be compared in terms of the number of instructions that need to provided to it? But I think for a successful ICLR paper, we would need 1-2 more meaningful baselines.

“Finally, we observe a virtuous circle that arises.” -> is there some mechanism to ensure that this is a virtuous cycle and not a vicious one? Couldn’t we just have horrible label corruption and then everything goes bad?

How easily would this scale to more temporally extended tasks in minigrid which have larger grids and more challenging tasks which are harder to solve in the sparse reward case?

Can we analyze whether the language goal space has some favorable generalization properties over a state based goal space as typical HER uses?

The language analysis in Section 4.4 is quite insightful and shows the good performance of the instruction generator over time.

How would this fare as language got more ambiguous and multimodal and the instruction generator had a harder time as well as HER might generalize more poorly?

Related Work
Fu et al (From Language to Goals: Inverse Reinforcement Learning for Vision-Based Instruction Following) might be relevant for instruction following as well, and some of Karthik Narasimhans work.

Learning interactively with language would also be related to Co-Reyes et al (Guiding Policies with Language via Meta-Learning)

Yiding Jiang’s recent work would also be relevant (https://arxiv.org/abs/1906.07343)


Overall I like the formulation, and it seems pretty useful for instruction following. But we need more comparisons, and a little more motivation on when this thing might become degenerate coz of the m labeling. Perhaps even a discussion/experiment on the uncertainty measure might be helpful.

**Experience Assessment:**

I have published one or two papers in this area.

**Review Assessment: Checking Correctness Of Derivations And Theory:**

N/A

**Review Assessment: Checking Correctness Of Experiments:**

I assessed the sensibility of the experiments.

**Review Assessment: Thoroughness In Paper Reading:**

I read the paper thoroughly.

---

> ### Author Response · Authors · 2019-11-14
> **Official Comment (2/2)**
>
>
> Baselines -> Thank you for raising this point. Although we design the experimental protocol to assess THER, we acknowledge that other baselines can give additional intuition, and we thus add another baseline (cf. Figure 4). As first suggested, we cannot benchmark Jiang et al. [8] models as they are tackling a slightly different problem: they perform *internal* instruction following operations inside the network, which is different from the UVFA setting that we are considering. Besides, the language models use pre-computed instructions, which cannot be generalized to unseen goals. The authors even mention that they leave instruction generator to future works, and "could not leverage the structure of language" (page 16).
> We also consider auxiliary losses or inductive neural biases; although complementary, we believe that they are too orthogonal to our approach to be relevant in the current paper.
> As a result, we designed a reward shaping baseline where the agent has a reward of 0.25 for every matching property in the instruction when it picks an object. This baseline enforces a hand-crafted curriculum and dramatically reduces the reward sparsity, which is the key property we want to evaluate. In the experiments, this strong baseline only slightly outperforms DQN+HER, and DQN+THER is only a few percentage points away. Yet, such reward shaping requires human expertise and may alter the optimal policy [9], while THER has a close score without these drawbacks. In total, we have three baselines DQN, DQN+reward\_shaping, and DQN+HER.
>
>
> Vicious Circle -> As mentioned earlier, we recommend to use a validation accuracy to train the mapper, and avoid over-fitting or reduced efficiency. As the mapper complement the Q-learner, it can be triggered anytime to kick-start the learning. It is thus more flexible than auxiliary losses, or inductive network biases that could also negatively impact the training (e.g., pixel prediction [1], hard-coded neural hierarchy - PACMAN [2]). We updated the Algorithm and pointed out this pipeline in the paper in the limitation paragraph.
>
>
> Temporally extended tasks -> As discussed with reviewer #1, we believe that THER can be applied to temporally extended tasks, e.g., Room2Room [9], Touchdown [11], etc. Yet, we also believe that such environments would require a full paper on their own to be correctly evaluated. Nonetheless, there have been some hints that THER could be successfully applied in practice by warm starting the instruction generator with human demonstrations: [5] and [6]. THER can also be coupled with intrinsic motivation methods [5,6] to deal with longer trajectories if we want to avoid human annotation.
>
>
> Language vs. state-based generalization. -> We believe that language has natural generalization properties, which are not present in state-space goal descriptors. For instance,
>  - Language goal-descriptors is more compact, are easily interpretable, and have compositionality properties which may be absent of the state-space goal descriptors as described in the introduction.
>  - Language goal-descriptors remove potential distractors from the state/observation space: an observation may contain objects that are irrelevant to the goal at hand.
>  - Language goal-descriptors allow for more complex goals that cannot be represented in the state-space: e.g., negation (pick a ball that is not blue), basic reasoning (pick the biggest ball), missing properties.
>  - Language goal-descriptors are agnostic to the input space modality, while state-space goal descriptors depend on the input (which may impact learning quality, reduce potential transfer learning, require more engineering, etc.). In robotics, the input may be 2D coordinates, 3D coordinates, or RGB inputs, whereas the goal remains the same. Finally, [7] tried to generate visual goals along trajectories (which is a state-based ), but the authors had to implement a complex GAN pipeline to obtain a state-based goal generator, and the mapper could not generalize well to unknown scenarios.
>
>
>  Related Work
> ----------------------
>
> As suggested, we update the related work section with:
>  - Fu, Justin, et al. "From language to goals: Inverse reinforcement learning for vision-based instruction following." arXiv preprint arXiv:1902.07742 (2019). -> we added a few notes on IRL for instruction following
>  - Jiang, Yiding, et al. "Language as an Abstraction for Hierarchical Deep Reinforcement Learning."
>  - Co-Reyes, John D., et al. "Guiding policies with language via meta-learning." arXiv preprint arXiv:1811.07882 (2018).
>
> Thank you very much for pointing out those relevant works.
>
>
>  Conclusion
> ----------------------
> We hope that we have correctly answered your questions, e.g., adding another baseline and preventing vicious circles. We remain open to discussion and to other changes toward improving paper quality. Again, we thank you for your extensive feedback.

---

> > ### Author Response · Authors · 2019-11-14
> > **References**
> >
> >
> > [1] Jaderberg, Max, et al. "Reinforcement learning with unsupervised auxiliary tasks." arXiv preprint arXiv:1611.05397 (2016).
> > [2] Chaplot, Devendra Singh, et al. "Embodied Multimodal Multitask Learning." arXiv preprint arXiv:1902.01385 (2019).
> > [3] Wang, Xin, et al. "Reinforced cross-modal matching and self-supervised imitation learning for vision-language navigation." Proceedings of the IEEE Conference on Computer Vision and Pattern Recognition. 2019.
> > [4] Fried, Daniel, et al. "Speaker-follower models for vision-and-language navigation." Advances in Neural Information Processing Systems. 2018.
> > [5] Bellemare, Marc, et al. "Unifying count-based exploration and intrinsic motivation." Advances in Neural Information Processing Systems. 2016.
> > [6] Haber, Nick, et al. "Learning to play with intrinsically-motivated, self-aware agents." Advances in Neural Information Processing Systems. 2018.
> > [7] Sahni, Himanshu, et al. "Visual Hindsight Experience Replay." arXiv preprint arXiv:1901.11529 (2019).
> > [8] Jiang, Yiding, et al. "Language as an Abstraction for Hierarchical Deep Reinforcement Learning." arXiv preprint arXiv:1906.07343 (2019).
> > [9] Ng, Andrew Y., Daishi Harada, and Stuart Russell. "Policy invariance under reward transformations: Theory and application to reward shaping." ICML. Vol. 99. 1999.
> > [10] Anderson, Peter, et al. "Vision-and-language navigation: Interpreting visually-grounded navigation instructions in real environments." Proceedings of the IEEE Conference on Computer Vision and Pattern Recognition. 2018.
> > [11] Chen, Howard, et al. "Touchdown: Natural language navigation and spatial reasoning in visual street environments." Proceedings of the IEEE Conference on Computer Vision and Pattern Recognition. 2019.

---

> ### Author Response · Authors · 2019-11-14
> **Official Comment (1/2)**
>
> First of all, thank you for reading the paper thoroughly and taking the time to review it.
>
> Introduction
> ----------------------
>
> First of all, thank you for your recommendations to improve the introduction readability, and we updated the paper accordingly.
>
> We kept the questions:  "are we eventually [...] and language acquisition?". At the beginning of the introduction, we wanted to emphasize that interleaving RL and language may either conflict or help the learning process in the general case. Although instruction following has its specific motivations ("you can communicate novel objectives to your agent"), it is also a natural test-bed to couple language and RL, and study this interconnection.
>
> As suggested, we motivated the use of language goal descriptors over state-space goal descriptors (as used with regular HER) by the following example: "More generally, language adds a level of semantics, which allows generating textual objective that could not be encoded with spatial observations as in regular HER, e.g., "fetch a ball that is not blue" or "pick any red object" ".
>
> We agree that the line "triggers [...] policy learning” is misleading, and we removed it.
>
>
> Experiments
> ----------------------
>
> Why noisy-HER work? -> An accuracy of 80 percent per attribute implies that an instruction is completely wrong 41\% of the time (1 - 0.8^4). This means that 59\% of the remaining time, at least 1 attribute is correct. As shown by the reward shaping experiment (described below), rewarding the agent when the picked object shares some attributes with the correct one triggers learning and allow the policy to outperform a random one. As few objects are present in the scene, all four attributes are not necessary to discriminate between objects (sometimes 1 or 2 are sufficient, see Fig.8 in the Appendix), allowing the policy to pick the correct object from time to time.
>
> The synthetic noisiness distribution vs. real distribution -> As you mentioned, the distribution differs between the noisy-HER experiments and the final experiments.  For instance, the final experiment can repeat words, discard a property, or contain different values of the same property. Yet, these cases do not occur in noisy-HER. It may be hard to compare both distributions rigorously, and we are not sure how informative it could be: we mostly design noisy-HER as a proof of concept before trying our approach.
>
> Negative feedback loop (+Rev#2) -> Thank you for your remark, and we added a new paragraph and experiments to assess this point. Besides, we also described more rigorously how to train the THER mapper to limit this potential risk in the paper.
> As you suggested, we may fear that activating a poor quality mapper could hurt the agent training, and even create a negative feedback loop that would favor degenerate policies. We first thus launched our experiments by applying the mapper from the beginning. We observe that the agent always manages to learn a valid policy, and it was pretty robust to invalid instructions in our setting.
> As it may differ in other environments, we also updated the paper to explicitly evaluate the mapper quality by using a validation accuracy and triggering it once a threshold is reached (cf. Algorithm in the Appendix). In practice, we can save a percentage of the positive trajectories, and iteratively create a validation dataset. Besides, we can use any language metric (BLEU, RED, SPiCE, METEOR, etc.) to replace the parser accuracy when dealing with natural language (We updated Section 4.2, second paragraph).
>
>
> Applying external expert HER -> Some environment, e.g., Room2Room [10], Touchdown [11], comes with a static dataset, and there are no instruction generators (or oracles). Therefore, it is not possible to apply HER, and a mapping function must be learned, as advocated in this paper.

---

### Official Review · AnonReviewer1 · 2019-10-23
**Official Blind Review #1**

**Rating:** 3

**Review:**

What is the specific question/problem tackled by the paper?
This paper tackles the problem of learning language-conditioned policies from reinforcement learning. Unlike most language-conditioned navigation work which relies on human demonstrations (e.g. in the room2room environment), this work only learns from the agent’s experience using a generalization of hindsight experience replay.

Method overview:
THER (textual HER) generalizes HER to cases where goals are not in the same space as states. To deal with this gap, THER learns a mapping from state space to goal space using successful trajectories. This mapping is then used to relabel unsuccessful trajectories with a guess of what goal was reached. This intuitive approach allows the text-conditioned agent to reach  40% at a 2D navigation task when conditioned on text such as “Pick the large red circle”.

Strengths:
The method is well motivated and would be useful. The ablations of showing how many successful trajectories are needed to learn the mapping (~1000-5000), how many time steps are needed to reach 1000 successes (~400k steps), and how accurate the mapping needs to be for HER to work (~80%) and thorough and easy to understand. This experimental completeness is itself a contribution.

Additionally, although the authors do not discuss this, this method is actually agnostic to the particular modality (e.g. text) of the goal space and could be used anytime the goal space differs from the state space.

Weaknesses:
The primary weakness of the paper is that the testbed environment and the textual goals are very simple. The “language” is just a list of up to 4 attributes describing the different objects and the control is simple navigation without any walls of visual variation. Additionally, the method requires accidentally getting successful trajectories early in training in order to train the mapping, and in this environment it is very easy to get successful trajectories.
I would interested in seeing how this method would work in the room2room environment (or some other more complex task). While it is unlikely to outperform the prior methods that use the human demonstrations, it would be useful to see how close THER can get to that performance and with how many environment steps. The advantage of this environment is that it has real human knowledge, and the textual goals are limited in number, making the experiment much more realistic (as humans are unlikely to sit next to an agent and generate infinitely many diverse textual goals).

A missing ablation in Figure 4 left is THER without waiting 400k steps before relabeling. In realistic scenarios, we would not be able to evaluate the mapper ahead of time to know when to start relabeling. How is performance affected is this knowledge is not available?

Overall, I lean to reject the paper in it’s current form, I believe this paper would be more impactful with experiments involving more language complexity or more policy complexity.

**Experience Assessment:**

I have published in this field for several years.

**Review Assessment: Checking Correctness Of Derivations And Theory:**

N/A

**Review Assessment: Checking Correctness Of Experiments:**

I carefully checked the experiments.

**Review Assessment: Thoroughness In Paper Reading:**

I read the paper at least twice and used my best judgement in assessing the paper.

---

> ### Author Response · Authors · 2019-11-14
> **Official Comment**
>
> First of all, thank you for reading the paper thoroughly and taking the time to review it!
>
> Strengths
> -----------------
>
> Modality Agnostic. -> As you mentioned, the method is agnostic to the goal modality. We did not highlight this point on purpose as two workshop papers already deal with a goal mapping function in vision [1] and language [2]. Thus, we did not want to claim our procedure as a novel contribution. However, we acknowledge that this paper formalizes the training pipeline independently of the modality. After reading your comment, we thus decided to mention this point in at the end of Section 3 and the conclusion. Thank you for emphasizing this point.
>
>
> Weaknesses
> ---------------------
>
> We acknowledge the environment's apparent simplicity. However, we voluntary set it up this way to evaluate the impact of our method. More precisely, we increase the environment size, the number of objects per room, and the number of attributes to break a heavily-tuned DQN. For instance, BabyAI only has two attributes (color and shape), which was too limited to assess language compositionality. However, we did not use walls as they mostly increase the exploration difficulty, which is not the core issue of the paper. In other words, we designed the environment to analyze THER properties while developing as much intuition as possible. Would you recommend to make this point more explicit?
>
>
> We agree that THER requires a small number of positive samples to work. However, even the best RL algorithm would not work if it does not have reward signals, i.e., positive trajectories! It is a standard RL setting to be able to collect some positive samples with a random policy. In this paper, we show that THER reduces the number of positive trajectories to kickstart a DQN agent.
>
>
> We also share your view on assessing THER on more challenging tasks in the long run, e.g., Room2Room [3], Touchdown [4], etc. They are a natural extension to BabyAI, and they open interesting research problems such as complex language instructions, photo-realistic perceptions, or relying on static datasets. Thus, we believe that such environments would require a full paper on their own to be correctly evaluated. Nonetheless, there have been some hints that THER could be successfully applied in practice: [5] and [6] successfully train an instruction generator in the Room2Room dataset with 21k instructions. We could then warm-start the instruction generator (and the policy) before finetuning the agent. We updated the related section to reflect those research directions.
> In the end, this paper focuses on working on a synthetic environment to have a good understanding of the algorithm mechanism before scaling up to more complex settings. Therefore, we believe that it would still be impactful in its current form.
>
>
> We thank you for suggesting an additional ablation in Figure 4; we thus assessed different mapper accuracy levels in the Appendix (Some experiments are still running, and the paper will be progressively updated). In a few words, there is little impact toward triggering the mapper early during the training, and THER is pretty robust to invalid early instructions in this setting. Thus, there is little insensitive to wait for a perfect mapper, as it only delays the training.
>
> As it may differ in different environments, we also updated the paper to explicitly evaluate the mapper quality by using a validation accuracy and triggering it once a threshold is reached (cf. Algorithm in the Appendix + footnote in Section 3). In practice, we can save a percentage of the positive trajectories, and iteratively create a validation dataset. Besides, we can use any language metric (BLEU, RED, SPiCE, METEOR, etc.) to replace the parser accuracy when dealing with natural language (We updated Section 4.2, second paragraph). We believe this procedure to be task-agnostic, rigorous, and easily scalable to complex scenarios.
>
>
> Conclusion
> -----------------
>
> Again, thank you very much for your comments, we hope to have answered some of your concerns such as the potential scalability of the issue and running an ablation study. We are open to discussion or additional changes if you think they can improve the paper further.

---

> > ### Author Response · Authors · 2019-11-14
> > **References**
> >
> >
> > [1] Sahni, Himanshu, et al. "Addressing Sample Complexity in Visual Tasks UsingHER and Hallucinatory GANs." arXiv preprint arXiv:1901.11529 (2019).
> > [2] Chan, Harris, et al. "ACTRCE: Augmenting Experience via Teacher's Advice For Multi-Goal Reinforcement Learning." arXiv preprint arXiv:1902.04546 (2019).
> > [3]Anderson, Peter, et al. "Vision-and-language navigation: Interpreting visually-grounded navigation instructions in real environments." Proceedings of the IEEE Conference on Computer Vision and Pattern Recognition. 2018.
> > [4] Chen, Howard, et al. "Touchdown: Natural language navigation and spatial reasoning in visual street environments." Proceedings of the IEEE Conference on Computer Vision and Pattern Recognition. 2019.
> > [5] Wang, Xin, et al. "Reinforced cross-modal matching and self-supervised imitation learning for vision-language navigation." Proceedings of the IEEE Conference on Computer Vision and Pattern Recognition. 2019.
> > [6] Fried, Daniel, et al. "Speaker-follower models for vision-and-language navigation." Advances in Neural Information Processing Systems. 2018.

---

### Author Response · Authors · 2019-11-14
**Paper Updates**

We want to thank the reviewers again for their comments and questions.

Following the reviewer feedback, we made the following updates to the paper:
 - Minor changes in the introduction following Reviewer 2 comments
 - Explicit a validation procedure in Section 3 and Appendix to avoid ill-trained goal instructor
 - Explicit that the accuracy metric can be replaced with language score, e.g., BLEU, RED, etc. in Section 4.2 and Appendix
 - Add a new baseline (reward shaping) in Figure 4.
 - Describe the baseline in section 4.3 - Baseline / Results.
 - Update Section 4.3 - Limitations:
     * Explicit HER vs. THER in the sparse reward setting
     * Mention potential issues with ill-trained generators
- Add Ablation study: when should we activate the instruction generator vs. the number of collected positive trajectories. Figure 9 in the Appendix. Note that other plots are running (3000 positive trajectories)
 - Add IRL for instruction following in the Related Work section
 - Add Conditioned Language Policy in the Related Work section
 - Mention that THER can be extended to other modality in the conclusion

We hope these responses satisfactorily address the raised concerns.

---

### Decision · Program_Chairs · 2019-12-19

**Decision:**

Reject

**Comment:**

Two reviewers are borderline and one recommends rejection. The main criticism is the simplicity of language, scalability to a more complex problem, and questions about experiments. Due to the lack of stronger support, the paper cannot be accepted at this point. The authors are encouraged to address the reviewer's comments and resubmit to a future conference.